

# Risk of lung cancer in patients with gastro-esophageal reflux disease: a population-based cohort study

Chi-Kuei Hsu[1], Chih-Cheng Lai[2], Kun Wang[3,*] and Likwang Chen[4,*]

[1] Department of Internal Medicine, E-Da Hospital, Kaohsiung, Taiwan
[2] Department of Intensive Care Medicine, Chi Mei Medical Center, Liouying, Tainan, Taiwan
[3] Department of Internal Medicine, Cardinal Tien Hospital, New Taipei City, Taiwan
[4] National Health Research Institutes, Miaoli, Taiwan
[*] These authors contributed equally to this work.

## ABSTRACT

This large-scale, controlled cohort study estimated the risks of lung cancer in patients with gastro-esophageal reflux disease (GERD) in Taiwan. We conducted this population-based study using data from the National Health Insurance Research Database of Taiwan during the period from 1997 to 2010. Patients with GERD were diagnosed using endoscopy, and controls were matched to patients with GERD at a ratio of 1:4. We identified 15,412 patients with GERD and 60,957 controls. Compared with the controls, the patients with GERD had higher rates of osteoporosis, diabetes mellitus, asthma, chronic obstructive pulmonary disease, pneumonia, bronchiectasis, depression, anxiety, hypertension, dyslipidemia, chronic liver disease, congestive heart failure, atrial fibrillation, stroke, chronic kidney disease, and coronary artery disease (all $P < .05$). A total of 85 patients had lung cancer among patients with GERD during the follow-up of 42,555 person-years, and the rate of lung cancer was 0.0020 per person-year. By contrast, 232 patients had lung cancer among patients without GERD during the follow-up of 175,319 person-years, and the rate of lung cancer was 0.0013 per person-year. By using stepwise Cox regression model, the overall incidence of lung cancer remained significantly higher in the patients with GERD than in the controls (hazard ratio, 1.53; 95% CI [1.19–1.98]). The cumulative incidence of lung cancer was higher in the patients with GERD than in the controls ($P = .0012$). In conclusion, our large population-based cohort study provides evidence that GERD may increase the risk of lung cancer in Asians.

Corresponding authors
Kun Wang, kunwang@mospital.com
Likwang Chen, likwang@nhri.org.tw

# INTRODUCTION

Gastro-esophageal reflux disease (GERD) is a condition that develops when the reflux of stomach contents causes troublesome symptoms and complications (*Bredenoord, Pandolfino & Smout, 2013*; *Moayyedi & Talley, 2006*; *Vakil et al., 2006*). GERD is a global issue that affects both children and adults (*El-Serag et al., 2014*). The incidence of the disease appears to have increased during the past 2 decades, particularly in North America and East Asia (*El-Serag, 2007*; *El-Serag et al., 2014*; *Vakil, 2010*). The most common manifestations

of GERD are esophageal symptoms, including heartburn, dysphagia, and regurgitation, and it can cause extra-esophageal presentation such as bronchospasm, laryngitis, and chronic cough. Because it may cause lung injury from recurrent microaspiration, GERD is associated with the risk of several lung diseases, such as idiopathic pulmonary fibrosis, cystic fibrosis, connective tissue disease, asthma, chronic obstructive pulmonary disease (COPD), and interstitial lung disease (*Blondeau et al., 2008*; *D'Ovidio et al., 2005*; *Mise et al., 2010*; *Morehead, 2009*; *Pacheco-Galvan, Hart & Morice, 2011*; *Pashinsky, Jaffin & Litle, 2009*; *Salvioli et al., 2006*; *Sweet et al., 2009*; *Patti et al., 2008*).

Recently, *Vereczkei et al. (2008)* investigated the association between GERD and non-small cell lung cancer (NSCLC) and found that a considerably higher proportion of patients with NSCLC had GERD than the general population, irrespective of cell type. Therefore, a study proposed that GERD-associated chronic lung injury may be one element of lung cancer promotion (*Herbella et al., 2015*). However, it enrolled only 25 patients with surgically treated adenocarcinoma and squamous cell carcinoma, and the relationship between GERD and lung cancer remains unclear (*Herbella et al., 2015*; *Vereczkei et al., 2008*). Therefore, whether GERD is associated with an increased risk of lung cancer should be determined. Hence, we performed a large-scale, controlled cohort study to estimate the hazard rates of lung cancer in patients with GERD by using a nationwide, population-based database in Taiwan.

## MATERIALS & METHODS

### Data source

The National Health Insurance (NHI) program of Taiwan is a nationwide insurance program that covers outpatient visits, hospital admissions, prescriptions, interventional procedures, and disease profiles for >99% of the population of Taiwan (23.12 million people in 2009) (*Chen et al., 2011*). Taiwan's National Health Research Institute (NHRI) used the original data from the NHI program to construct a longitudinal database of patients admitted between 1997 and 2010. This cohort includes 2,619,534 hospitalized patients, representing 10% of all NHI enrollees. This sampled fraction (a 3.4:1 ratio) is based on a regulation that limits the maximal amount of NHI data that can be extracted for research purposes. The National Health Insurance Research Database (NHIRD) is one of the largest and most comprehensive databases worldwide and has been used extensively in various studies of prescription use, diagnoses, and hospitalizations. This study was approved by the Institutional Review Board of Cardinal Tien Hospital (Number: EC1011008-E-R1).

### Identification of patients with GERD and without GERD

To investigate the associations between GERD and the risk of lung cancer, we performed a cohort study. All beneficiaries with GERD from 1997 to 2010 were extracted using the International Classification of Diseases, Ninth Revision, Clinical Modification (ICD-9-CM) codes 530.85, 530.11, and 530.81. Patients with GERD were identified using the ICD-9-CM codes and procedure codes for endoscopy as previous study (*Lee et al., 2014*). Patients who were not diagnosed with GERD after receiving endoscopy were excluded. Patients with a history of lung cancer or peptic ulcer disease were also excluded. We matched controls (patients

without GERD) to patients with GERD by age, sex, and the index date at a ratio of 1:4. In the non-GERD group, patients with a history of lung cancer or peptic ulcer disease were excluded.

## Baseline variables

We collected data on demographic and clinical characteristics of the study population, including age, sex, and comorbidities. Comorbidities were defined according to the ICD-9-CM and procedure codes within 1 year before index admission. We used a relatively strict criterion to define comorbidities: coding one morbidity required at least one admission or 3 outpatient clinic visits for disease treatment during the year before index admission.

## Definition of outcome

We followed up each patient until December 31, 2010, to observe for the development of de novo lung cancer. In Taiwan, patients with cancer can apply for a catastrophic illness certificate that exempts them from any out-of-pocket expenses for cancer evaluation and care. The development of de novo lung cancer was identified by the ICD-9-CM code 162 having been noted on the catastrophic illness certificate as a previous study (*Jian et al., 2015*). The follow-up duration was calculated from the date of GERD diagnosis (index date) to the date of the first recorded cancer code.

## Statistical analysis

All data were analyzed using SAS Version 9.3 software (SAS Institute). Categorical variables are expressed as numbers or percentages and were compared using the chi-square test. Incidence rates of lung cancer in both GERD and non-GERD groups were calculated by Poisson regression. The Kaplan–Meier method was used to estimate the cumulative incidence rate of lung cancer in patients with or without GERD. The cumulative incidence curves of both groups were compared using the log-rank test. We used Log-Minus-Log survival plots to evaluate proportional hazard assumption. To assess the risk of lung cancer, a list of potential risk factors associated both with admission of lung cancer and with GERD status was considered in the Cox regression model. Univariable and multivariable Cox regression models (stepwise selection) were performed to examine the association of lung cancer with potential confounding factors such as osteoporosis, diabetes mellitus (DM), asthma, COPD, pneumonia, anxiety, hypertension, dyslipidemia, chronic liver disease, congestive heart failure (CHF), atrial fibrillation, stroke, chronic kidney disease (CKD), and coronary artery disease (CAD). Two-sided $P$ values < .05 were considered statistically significant.

## RESULTS

Initially, the NHIRD was used to identify 97,221 patients diagnosed with GERD after undergoing esophagogastroduodenoscopy from January 1, 1997 to December 31, 2010. After excluding 943 patients aged <18 or >100 years, 71,255 patients diagnosed with lung cancer or peptic ulcer before the index date, 9,397 patients without GERD diagnosis 1 year later after the index date, and 182 patients with missing demographic data, we found only 15,444 patients eligible for matching. Overall, we identified 15,444 patients with GERD and 60,957 age- and sex-matched controls (Fig. 1).

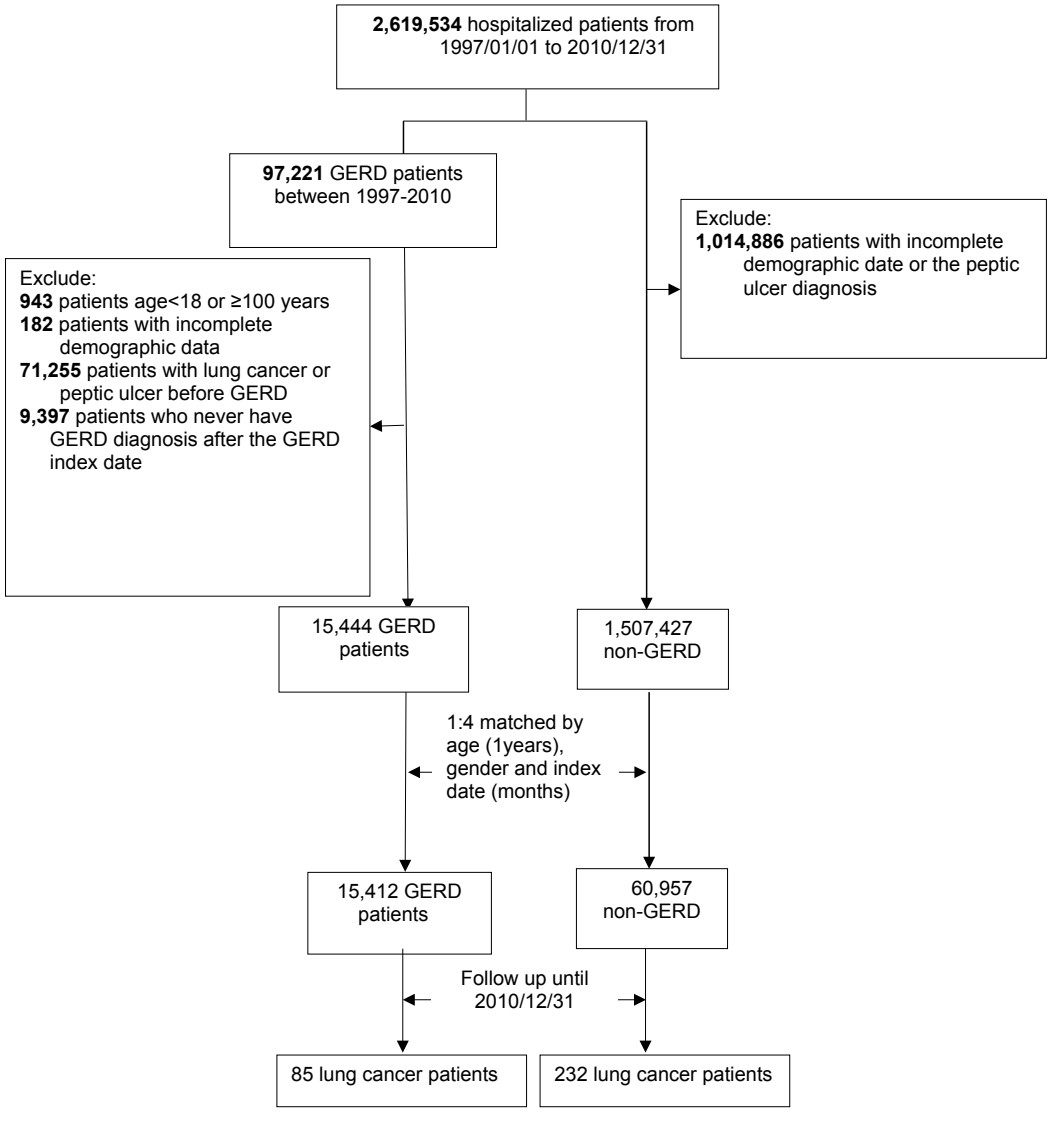

**Figure 1  Study algorithm for patient enrollment.**

A total of 85 patients had lung cancer among patients with GERD during the follow-up of 42,555 person-years, and the rate of lung cancer was 0.0020 per person-year. By contrast, 232 patients without GERD had lung cancer during the follow-up of 175,319 person-years, and the rate of lung cancer was 0.0013 per person-year (Table 1). The baseline characteristics and comorbidities are listed in Table 2. Compared with the controls, the patients with GERD displayed higher rates of osteoporosis, asthma, COPD, pneumonia, bronchiectasis, depression, anxiety, hypertension, dyslipidemia, chronic liver disease, CHF, atrial fibrillation, stroke, CKD, and CAD (all $P < .05$). There is no violation of the proportional hazard assumption. By using stepwise Cox regression model, we found all potential confounding variables are not significantly associated with lung cancer except for GERD (HR, 1.53; 95%

Table 1 Incidence rates of lung cancer events per 10,000 person-year among gastro-esophageal reflux disease (GERD) and non-GERD group.

| Groups | Person-year | Number of lung cancer | Rate (per 10,000 person-year) | 95% CI |
|---|---|---|---|---|
| GERD | 42555.51 | 85 | 0.0020 | (0.0016, 0.0024) |
| nonGERD | 175319.72 | 232 | 0.0013 | (0.0012, 0.0015) |

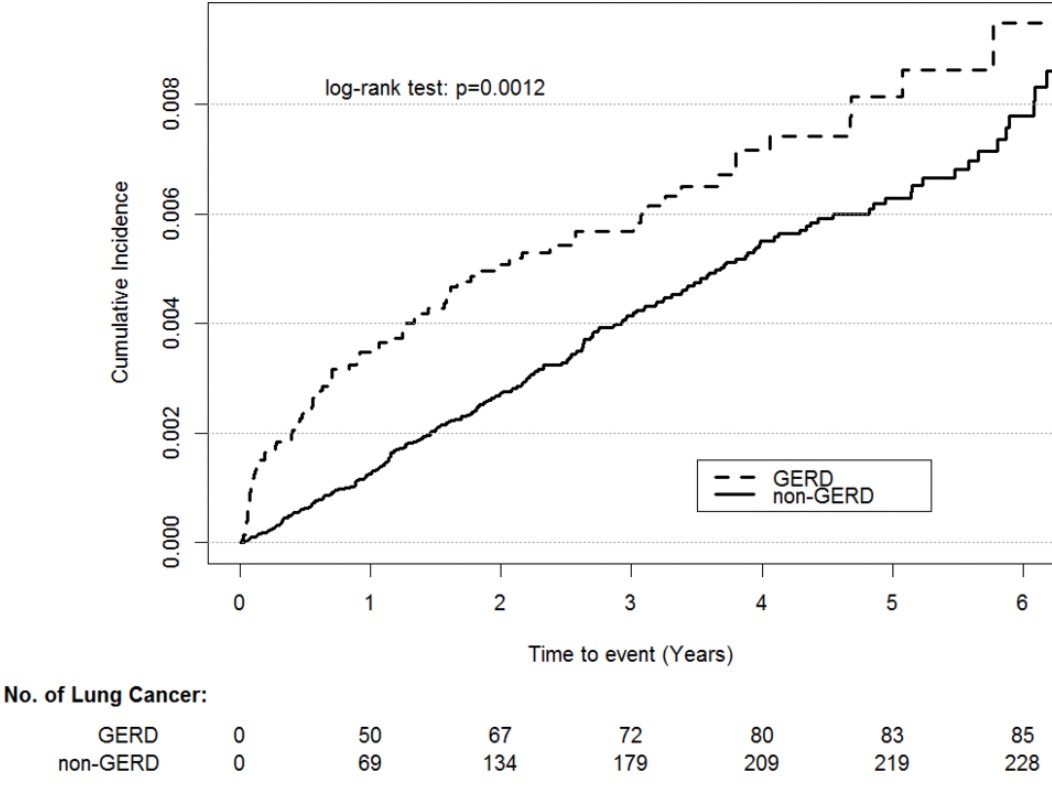

**No. of Lung Cancer:**

| | | | | | | | |
|---|---|---|---|---|---|---|---|
| GERD | 0 | 50 | 67 | 72 | 80 | 83 | 85 |
| non-GERD | 0 | 69 | 134 | 179 | 209 | 219 | 228 |

Figure 2 Cumulative incidence rate of lung cancer for patients with or without GERD.

CI [1.19–1.98]; Table 3). As shown in Fig. 2, the cumulative incidence of lung cancer was higher in the patients with GERD than in the controls ($P = .0012$).

## DISCUSSION

This large, population-based, long-term follow-up cohort study is the first to investigate the relationship between GERD and lung cancer. Besides the strong association between GERD and esophageal cancer, several studies have shown that GERD is also an important risk factor for laryngeal/pharyngeal cancer (*Bacciu et al., 2004*; *Langevin et al., 2013*; *Vaezi et al., 2006*). Additionally, the significant association between GERD and laryngeal cancer with pooled odds ratios of 2.86 (95% CI [2.73–2.99]) and 2.37 (95% CI [1.38–4.08]) on the basis of fixed-effect and random-effect models, respectively, were demonstrated in one meta-analysis (*Qadeer, Colabianchi & Vaezi, 2005*). Although it may be logical that the lungs, as one of the organs near the esophagus, and should be affected by the gastric

**Table 2** Baseline characteristics of study population stratified by gastro-esophageal reflux disease (GERD) and non-GERD group.

| Variables | Number (%) of patients with GERD $N = 15,412$ | Number (%) of patients without GERD $N = 60,957$ | $\chi^2\,(df)$ | $p$-value |
|---|---|---|---|---|
| **Gender** | | | 0.326(1) | 0.568 |
| Female | 7,849 (50.9) | 31,201 (51.2) | | |
| Male | 7,563 (49.1) | 29,756 (48.8) | | |
| **Age group** | | | 4.203(2) | 0.122 |
| 18–54 years | 6,503 (42.2) | 26,049 (42.7) | | |
| 54–64 | 5,961 (38.7) | 23,677 (38.8) | | |
| ≥65 years | 2,948 (19.1) | 11,231 (18.4) | | |
| **Underlying diseases/conditions** | | | | |
| Osteoporosis | | | 15.366(1) | <.001 |
| No | 15,173 (98.5) | 60,250 (98.84) | | |
| Yes | 239 (1.6) | 707 (1.2) | | |
| Diabetes mellitus | | | 16.298(1) | <.001 |
| No | 14,078 (91.3) | 55,030 (90.3) | | |
| Yes | 1,334 (8.7) | 5,927 (9.7) | | |
| Tuberculosis | | | 0.930(1) | 0.335 |
| No | 15,342 (99.6) | 60,714 (99.6) | | |
| Yes | 70 (0.5) | 243 (0.4) | | |
| Asthma | | | 95.058(1) | <.001 |
| No | 14,881 (96.6) | 59,673 (97.9) | | |
| Yes | 531 (3.5) | 1,284 (2.1) | | |
| Chronic obstructive pulmonary diseases | | | 259.590(1) | <.001 |
| No | 14,158 (91.9) | 58,015 (95.2) | | |
| Yes | 1,254 (8.4) | 2,942 (4.8) | | |
| Pneumonia | | | 247.106(1) | <.001 |
| No | 14,929 (96.9) | 60,157 (98.7) | | |
| Yes | 483 (3.1) | 800 (1.3) | | |
| Pneumoconiosis | | | 0.031(1) | 0.861 |
| No | 15,401 (99.9) | 60,916 (99.9) | | |
| Yes | 11 (0.1) | 41 (0.1) | | |
| Bronchiectasis | | | 28.373(1) | <.001 |
| No | 15,349 (99.6) | 60,847 (99.8) | | |
| Yes | 63 (0.4) | 110 (0.2) | | |
| Depression | | | 37.239(1) | <.001 |
| No | 15,242 (98.9) | 60,570 (99.4) | | |
| Yes | 170 (1.1) | 387 (0.6) | | |
| Anxiety | | | 365.826(1) | <.001 |
| No | 14,167 (91.9) | 58,337 (95.7) | | |
| Yes | 1,245 (8.1) | 2,620 (4.3) | | |
| Hypertension | | | 9.954(1) | 0.002 |
| No | 12,031 (78.1) | 48,291 (79.2) | | |
| Yes | 3,381 (22.0) | 12,666 (20.8) | | |

**Table 2** (*continued*)

| Variables | Number (%) of patients with GERD $N = 15,412$ | Number (%) of patients without GERD $N = 60,957$ | $\chi^2(df)$ | *p*-value |
|---|---|---|---|---|
| Dyslipidemia | | | 11.772(1) | <.001 |
| No | 13,993 (90.8) | 55,871 (91.7) | | |
| Yes | 1,419 (9.2) | 5,086 (8.3) | | |
| Chronic liver disease | | | 75.039(1) | <.001 |
| No | 14,865 (96.5) | 59,546 (97.7) | | |
| Yes | 547 (3.6) | 1,411 (2.3) | | |
| Congestive heart failure | | | 21.013(1) | <.001 |
| No | 15,118 (98.1) | 60,101 (98.6) | | |
| Yes | 294 (2.0) | 856 (1.4) | | |
| Atrial fibrillation | | | 4.635(1) | 0.031 |
| No | 15,274 (99.1) | 60,514 (99.3) | | |
| Yes | 138 (0.9) | 443 (0.7) | | |
| Myocardial infarction | | | 0.289(1) | 0.591 |
| No | 15,316 (99.4) | 60,600 (99.4) | | |
| Yes | 96 (0.62) | 357 (0.6) | | |
| Stroke | | | 19.921(1) | <.001 |
| No | 14,581 (94.6) | 58,190 (95.5) | | |
| Yes | 831 (5.4) | 2,767 (4.6) | | |
| Peripheral vascular disease | | | 0.720(1) | 0.720 |
| No | 15,350 (99.6) | 60,724 (99.6) | | |
| Yes | 62 (0.4) | 233 (0.7) | | |
| Chronic kidney diseases | | | 32.318(1) | <.001 |
| No | 15,083 (97.9) | 60,050 (98.5) | | |
| Yes | 329 (2.1) | 907 (1.5) | | |
| Coronary artery diseases | | | 32.782(1) | <.001 |
| No | 15,051 (97.7) | 59,947 (98.3) | | |
| Yes | 361 (2.3) | 1,010 (1.7) | | |

refluxate, no study has assessed the possible relationship between GERD and lung cancer. Our study is the first to demonstrate a significant positive association between GERD and lung cancer. This finding was supported by the increased risk of lung cancer in comparison with age- and sex-matched controls (crude HR, 1.53; 95% CI [1.19–1.98]). Our findings have some clinical implications. After confirming this significant association between GERD and lung cancer, it was suggested that aggressive treatment of GERD possibly prevents the development of lung cancer. However, further studies should be warranted to prove the possible chemopreventive role of antacid use in patients with GERD.

Our study has several strengths. First, all of the patients with GERD and controls in this study were enrolled from the Taiwan NHIRD, which is a highly representative database. Therefore, the bias of recall and selection can be minimized. Second, our study identified lung cancer patients by using valid and definite approaches. In the Taiwan NHI program, individuals with registration of cancer for a catastrophic illness certificate required biopsy and histological verification. Third, by using medical records from NHIRD, we can reduce

**Table 3  Crude hazard ratios (HR) among gastro-esophageal reflux disease (GERD) and non-GERD group.**

| Variables | Beta value | Crude HR (95%CI) | p value |
| --- | --- | --- | --- |
| GERD | 0.43 | 1.53 (1.19–1.98) | 0.001 |
| Osteoporosis | 0.59 | 1.80 (0.88–3.69) | 0.110 |
| Diabetes mellitus | 0.05 | 1.05 (0.76–1.46) | 0.752 |
| Asthma | 0.20 | 1.22 (0.65–2.29) | 0.538 |
| Chronic obstructive pulmonary diseases | 0.30 | 1.34 (0.92–1.96) | 0.125 |
| Pneumonia | 0.40 | 1.49 (0.76–2.90) | 0.245 |
| Bronchiectasis | 0.69 | 2.00 (0.18–22.06) | 0.571 |
| Depression | 0.84 | 2.31 (0.55–9.69) | 0.251 |
| Anxiety | −0.08 | 0.92 (0.53–1.60) | 0.765 |
| Hypertension | −0.11 | 0.90 (0.69–1.17) | 0.429 |
| Dyslipidemia | −0.02 | 0.98 (0.65–1.46) | 0.913 |
| Chronic liver disease | −0.25 | 0.78 (0.36–1.71) | 0.539 |
| Congestive heart failure | −0.88 | 0.42 (0.15–1.19) | 0.102 |
| Atrial fibrillation | −1.23 | 0.29 (0.07–1.25) | 0.098 |
| Stroke | −0.42 | 0.66 (0.41–1.05) | 0.079 |
| Chronic kidney diseases | −0.31 | 0.73 (0.33–1.60) | 0.433 |
| Coronary artery diseases | −0.28 | 0.76 (0.36–1.58) | 0.458 |

the likelihood of non-response and loss of follow-up to a minimum. Besides, there were some variables during the multivariable analysis. We controlled them by statistic methods (Table 3). Most important of all, we used a nationwide and population-based database–Taiwan NHIRD. Thus, the findings in the present work can be generalized in the real world.

Several mechanisms can help explain the significant relationship between GERD and lung cancer. First, several studies have shown that the refluxate can destroy the epithelium of the larynx or pharynx by means of introducing chronic inflammation (*Rees et al., 2008*) or activating proliferative signaling pathways (*Dvorak et al., 2011*; *Johnston et al., 2012*; *Sung et al., 2003*) and further result in malignant transformation. In addition, based on the studies investigating the pathogenesis of Barrett's esophagus and esophageal carcinoma, both acid and bile can promate carcinogenesis through the induction of DNA damage and the influence of cell proliferation and apoptosis (*Denlinger & Thompson, 2012*; *Fang et al., 2013*). These pathogenesis may happen in the respiratory tract, and contribute to the development of lung cancer. Second, the trend of the predominance of lung adeno-carcinoma among all cell type is similar with the distribution trend of esophageal cancer (*Etzel et al., 2006*; *Liam et al., 2006*). Third, the origin of central lung adenocarcinoma is different from peripheral lung cancer's. Lung cancer at a central site is more prone to be affected by gastric refluxate than at a peripheral site. Thus, lung adenocarcinoma at a central site is more likely to arise in the glandular epithelium in contrast to lung cancer at a peripheral site which possibly originates from type II pneumocytes and Clara cells (*Fukui et al., 2013*).

However, this study also had several limitations. First, we could not obtain data such as smoking, which is an important risk factor for both GERD and lung cancer. However, we tried to include some smoking-related disorders such as dyslipidemia, hypertension,

CAD, or COPD to minimize the influence of smoking. In addition, the data regarding the type of lung cancer was not available. Therefore, we cannot further analyze the association between GERD and the specific type of lung cancer. Second, patients with GERD may more often visit physicians than patients without GERD and this difference may cause possible surveillance bias. Finally, we did not collect the data about the use of anti-GERD treatments such as proton pump inhibitors or histamine-2-receptor antagonist.

## CONCLUSIONS

Our large, population-based cohort study provides evidence that GERD may increase the risk of lung cancer.

### Funding
This study was supported by grants from National Health Research Institutes (intramural funding). The funders had no role in study design, data collection and analysis, decision to publish, or preparation of the manuscript.

### Grant Disclosures
The following grant information was disclosed by the authors:
National Health Research Institutes.

### Competing Interests
The authors declare there are no competing interests.

### Author Contributions
- Chi-Kuei Hsu and Chih-Cheng Lai conceived and designed the experiments, wrote the paper.
- Kun Wang prepared figures and/or tables, reviewed drafts of the paper.
- Likwang Chen analyzed the data, prepared figures and/or tables, reviewed drafts of the paper.

### Ethics
The following information was supplied relating to ethical approvals (i.e., approving body and any reference numbers):
    Cardinal Tien Hospital (Approval number: EC1011008-E-R1).

### Data Availability
    The raw data for this work was obtained by application from the National Health Insurance Research Database, Taiwan (http://nhird.nhri.org.tw/en/) and may not be shared according to the Database's rules governing use. Access to the data used in this study may be obtained by citizens of the Republic of China who fulfill the requirements of conducting research projects.

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
