# Peer review of "Risk of lung cancer in patients with gastro-esophageal reflux disease: a population-based cohort study"

_PeerJ, doi:10.7717/peerj.2753_

## Round 0.1 · original submission · Minor Revisions

As a whole, the reviewers did not raise any major concerns. Please address the comments given by respective reviewers and I look forward to your revised version.

·

Basic reporting

No comments.

Experimental design

No comments.

Validity of the findings

No comments.

Additional comments

General comments
1. Types of lung cancers found among subjects with gastro-esophageal reflux disease and controls needs to be mentioned.

Reviewer 2 ·

Basic reporting

No comments

Experimental design

No comments

Validity of the findings

No comments

Additional comments

- In Materials & Methods, for section Statistical analysis, authors should explain Kaplan-Meier method before survival anlaysis by using Cox regression models.
- Line 108: “multivariate survival analysis” should be “multivariable survival analysis”. “Multivariate” means you have more than one dependent variables. But in your analysis you have one dependent variable (i.e., lung cancer)
- Table 1: standardise the p-value decimal to three
- Standardise the format of p-value, eg. <0.001 NOT <.001
- Authors should indicate what method is used in selecting the significant variables in the multiple cox regression (e.g., forward stepwise)
- In Materials & Methods, for section Statistical analysis, authors should indicate how the assumptions of the survival analysis (multiple cox regression) were checked, how the model fitness was assessed.
- The detail of the survival analysis using simple and multiple cox regression result should be presented in a table. All significant confounding variables (age, sex etc) should be included in the Table.
- The authors should report the simple cox regression (b value, crude HR(95%CI), p-value) and multiple cox regression (b value, Adjusted HR(95%CI) and p-value). Table 2 is too brief for the study.

·

Basic reporting

Properly written paper on an interesting topic - ie. correlation between GERD and NSCLC - causative or synchronous events due to a (common?) third factor - unknown so far..

The original hypothesis was laid down by me, as last author - so I see it as my brain child.
I am more than happy to read a really good research on the topic - worth to publish.

Experimental design

Proper study design - concrete limits.
The mathematic used is proportional and well choosen.

Data are reliable - properly collected, controlled.

Validity of the findings

The finidngs are valid.

The explanations are convincing - and I like, it very much,how the authors limit
their own observations - stay away from generalisation or overexpression.

Yes, the final question remains open - is it possible, that smoking is a common
contributor to both pathologies? But we need such publications to trigger further
studies.

Congratulation to the authors for their performance.

Additional comments

Congratulations to the authors for their performance: this is a good paper,
they grabbed a question, disturbing enough and gave a possible answer,
open enough to generate further studies.

Reviewer 4 ·

Basic reporting

Clear, unambiguous, professional English language used throughout.
Intro & background show context.
Literature well referenced & relevant. But should add on discussion since the authors need to control few variables during the multivariable analysis.
Should mention is there any loss to follow-up. Should revise the term incidence rate based on study design. Should mention all confounding covariates that were consider and significant during multiple cox regression. The title of the figure should be below.
Figure 1: Should be completed until the outcome measures
Figure 2: Should revise the title…incidence rate based on study design
Table 1: Should put n(%) in the heading. Usually we use 1 decimal for %, age group should be 18-54, 55, 64, >64. Should put value of 19 for bronchiectasis and statistical test used under footnotes. I think it is more relevance if the authors put the comorbid of lung disease or GERD not to put all the underlying disease and condition. The table with column those with and without lung cancer is more relevance. The few variables were significant, which may affect the results.

Experimental design

Methods should be described with sufficient detail & information to replicate.

Study design:
Should explain more specific regarding study design either retrospective or prospective cohort study or matched case control. The authors explained that GERD patients were matched with control (page 2), based on age, gender and index date (page 17). Title in Table 1 also mention regarding study population stratified by GERD and non-GERD group. Should state the rationale for matched samples. I think all these factors can be controlled during multivariable analysis e.g multiple cox regression. If case control study, the cases must be those with lung cancer and control that those without lung cancer. We will determine the risk of by looking how many have GERD in those with and without lung cancer. The authors reported on incidence which usually reported in prospective cohort study. In page 9 state long-term follow-up cohort study which mean prospective cohort study.

Study duration:
Should mention the duration of the study.

Study population:
Sample size: is it the correct reason for the ratio 3.4:1 is based on a regulation. It should be the ratio according to study design; for independent prospective studies m is the ratio of control to cases subjects. For matched prospective studies m is the number of controls subjects matched to each case subject. For independent case-control studies m is the ratio of control to case patients. For matched case-control studies m is the number of control patients matched to each case.
Inclusion exclusion criteria: the age for inclusion were stated differently; <40 or 100 years (page 8) but <18 0r >100 years (page 17). Should mention incomplete data as exclusion criteria and how much of incomplete data will be considered.

Data collection:
The authors should describe more detail regarding data collection procedure especially how to get the results of outcome. It just tracing the records or meeting with the patients. Should describe the measurement tool used.

Operational definition:
Should define the GERD and without GERD, comorbid of lung cancer or GERD, value for person-years. Should state or put reference for the ICD-9-CM codes for GERD and procedure codes for endoscopy. Ideally should put comorbid of GERD and lung cancer. The authors state comorbidities based on ICD-9-CM. Is there any reason why coding one morbidity required at least one admission or 3 outpatients clinic visits. Should state or put reference for the ICD-9-CM code162 for the development of de nove lung cancer.

Statistical analysis:
No reported relative risks, but the authors did mention it in page 4. Under survival analysis, should state the follow-up time for each of the patients and for those at the end of the study (31 December 2010), censored outcome.

Validity of the findings

Results:
Should mention is there any loss to follow-up. Should revise the term incidence rate based on study design. Should mention all confounding covariates that were consider and significant during multiple cox regression. The title of the figure should be below.

Additional comments

Methods:
Study design:
Should explain more specific regarding study design either retrospective or prospective cohort study or matched case control. The authors explained that GERD patients were matched with control (page 2), based on age, gender and index date (page 17). Title in Table 1 also mention regarding study population stratified by GERD and non-GERD group. Should state the rationale for matched samples. I think all these factors can be controlled during multivariable analysis e.g multiple cox regression. If case control study, the cases must be those with lung cancer and control that those without lung cancer. We will determine the risk of by looking how many have GERD in those with and without lung cancer. The authors reported on incidence which usually reported in prospective cohort study. In page 9 state long-term follow-up cohort study which mean prospective cohort study.

Study duration:
Should mention the duration of the study.

Study population:
Sample size: is it the correct reason for the ratio 3.4:1 is based on a regulation. It should be the ratio according to study design; for independent prospective studies m is the ratio of control to cases subjects. For matched prospective studies m is the number of controls subjects matched to each case subject. For independent case-control studies m is the ratio of control to case patients. For matched case-control studies m is the number of control patients matched to each case.
Inclusion exclusion criteria: the age for inclusion were stated differently; <40 or 100 years (page 8) but <18 0r >100 years (page 17). Should mention incomplete data as exclusion criteria and how much of incomplete data will be considered.

Data collection:
The authors should describe more detail regarding data collection procedure especially how to get the results of outcome. It just tracing the records or meeting with the patients. Should describe the measurement tool used.

Operational definition:
Should define the GERD and without GERD, comorbid of lung cancer or GERD, value for person-years. Should state or put reference for the ICD-9-CM codes for GERD and procedure codes for endoscopy. Ideally should put comorbid of GERD and lung cancer. The authors state comorbidities based on ICD-9-CM. Is there any reason why coding one morbidity required at least one admission or 3 outpatients clinic visits. Should state or put reference for the ICD-9-CM code162 for the development of de nove lung cancer.

Statistical analysis:
No reported relative risks, but the authors did mention it in page 4. Under survival analysis, should state the follow-up time for each of the patients and for those at the end of the study (31 December 2010), censored outcome.

Results:
Should mention is there any loss to follow-up. Should revise the term incidence rate based on study design. Should mention all confounding covariates that were consider and significant during multiple cox regression. The title of the figure should be below.
Figure 1: Should be completed until the outcome measures
Figure 2: Should revise the title…incidence rate based on study design
Table 1: Should put n(%) in the heading. Usually we use 1 decimal for %, age group should be 18-54, 55, 64, >64. Should put value of 19 for bronchiectasis and statistical test used under footnotes. I think it is more relevance if the authors put the comorbid of lung disease or GERD not to put all the underlying disease and condition. The table with column those with and without lung cancer is more relevance. The few variables were significant, which may affect the results.

Discussion:
Should add on discussion since the authors need to control few variables during the multivariable analysis.

---

## Round 0.2 · Minor Revisions

There are further concerns from reviewers that will require revisions. I hope the authors will look through them carefully. I look forward to a revised version.

·

Basic reporting

No comments

Experimental design

No comments

Validity of the findings

No comments

Reviewer 2 ·

Basic reporting

Acceptable

Experimental design

Acceptable

Validity of the findings

Acceptable

Additional comments

1) Line 114-116, if authors did not report the t-test result for comparison between 2 groups in numerical variable in this manuscript, then the statement of “Continuous variables are expressed as mean ± standard deviations and were compared using the Student t test.” should be removed from the text. For your information, Median (IQR) should be reported in Survival analysis NOT Mean (SD). Survival analysis that you used in this study is a non-parametric test, thus, median (IQR) should be reported for numerical variables.

2) Line 124, wrong spelling “Univairate”, should be “Univariable”

3) Line 142 -146. The result reported in this sentence is not consistent with Table 2. For example, “patient with GERD displayed higher rates of DM” whereas in Table 2, DM had 8.7%(with GERD) vs 9.7% (without GERD).

4) Line 252 and 256, check the references arrangement, should it be Denlinger… followed by D’Ovidio?

5) Table 2: The information provided in Table 2 is incomplete. I suggest the authors to provide number(%) for the comparison group. For example, provide number (%) for female, Osteoporosis (yes & no) etc… There is one missing bracket “Tuberculosis… 70(0.5)”. I suggest authors to state the statistical analysis used for the p-value in Table 2 by adding the X2(df) just before each p-value. Authors can add additional column for X2(df) before the p-value column.

6) Table 3: Independent variables (your confounding variables) that are not significant in your last model can be excluded from the model. Based on this table, only GERD was significant independent variable. Therefore, you may remove Table 3 from your manuscript, but report the crude HR, p-value of GERD in text. I hope you have used appropriate independent variables selection method during the analysis, and it should be end up with one significant independent variable (i.e., GERD). If all your confounding variables are not significant in the model, you should remove them from the final model analysis. Run the analysis using GERD (independent variable) and lung cancer (outcome variable) only in your final cox regression model. Explain clearly in text that all confounding variables are not significant associated with lung cancer except for GERD, then present the HR (95%CI) and its p-value for GERD.

Reviewer 4 ·

Basic reporting

No comments

Experimental design

No comments

Validity of the findings

No comments

Additional comments

Method
Study design should be the prospective cohort study. Not just mention cohort. The cohort study have retrospective, prospective and historical cohort. If prospective cohort, it is correct to report the incidence but not for retrospective cohort.

Should mention is there any loss to follow-up. The authors explain the loss of follow-up can be reduced to a minimum. The reason why the reviewer asks is that the loss of follow-up cases should be under censored observation. Those who do not have lung cancer also were considered under censored observation. This censored observation is an advantage of using survival analysis. It can underestimate of survival time of lung cancer if the loss to follow cases were excluded.

Duration of follow up? Is it enough to see the development of de novo lung cancer within three years? What about those diagnosed at the end of the study? Is there any follow up added to these groups of patients?

Sample size
The ratio of 3.4: 1 was based on the NHIRD database research regulation. It is true based on regulation. But for sample size determination for independent prospective studies m is the ratio of control to cases subjects based on the selected population.

Results
Authors should indicate what method is used in selecting the significant variables in the multiple cox regression (e.g., forward stepwise). The authors did not response to it.

Why the authors reported “adjusted for” as a footnote under Table 3. The reviewer thought that the factors were significant. All the factors mentioned were not significant even in simple cox regression. There was no different in the crude and adjusted hazard ratio. The results were taken from which method? The issue also was asked by another reviewer: Authors should indicate what method is used in selecting the significant variables in the multiple Cox regression (e.g., forward stepwise). The authors did not response to it. No need to report the non-significant factors. It was related to the comment on the comorbid of lung disease or GERD. Not to put all the underlying disease and condition. Usually, we selected those factors with p-value <0.25 or clinically important for variable selection in multiple Cox regression. There six factors with p-value <0.25. Others factors were also included. Are they clinically important confounder for the development of lung cancer?

Figure 2. delete “based on study design”

Table 3: standardise the p-value decimal to three

---

## Round 0.3 · accepted · Accept

The revised version has been reviewed and concerns addressed.

Reviewer 2 ·

Basic reporting

acceptable

Experimental design

acceptable

Validity of the findings

acceptable

Additional comments

Thanks for the corrections and improvement.